# Measuring Mathematical Problem Solving With the MATH Dataset

**Dan Hendrycks**
UC Berkeley

**Collin Burns**
UC Berkeley

**Saurav Kadavath**
UC Berkeley

**Akul Arora**
UC Berkeley

**Steven Basart**
UChicago

**Eric Tang**
UC Berkeley

**Dawn Song**
UC Berkeley

**Jacob Steinhardt**
UC Berkeley

## Abstract

Many intellectual endeavors require mathematical problem solving, but this skill remains beyond the capabilities of computers. To measure this ability in machine learning models, we introduce MATH, a new dataset of 12,500 challenging competition mathematics problems. Each problem in MATH has a full step-by-step solution which can be used to teach models to generate answer derivations and explanations. To facilitate future research and increase accuracy on MATH, we also contribute a large auxiliary pretraining dataset which helps teach models the fundamentals of mathematics. Even though we are able to increase accuracy on MATH, our results show that accuracy remains relatively low, even with enormous Transformer models. Moreover, we find that simply increasing budgets and model parameter counts will be impractical for achieving strong mathematical reasoning if scaling trends continue. While scaling Transformers is automatically solving most other text-based tasks, scaling is not currently solving MATH. To have more traction on mathematical problem solving we will likely need new algorithmic advancements from the broader research community.

## 1  Introduction

Mathematics is a highly effective tool in many intellectual endeavors. It enables us to count and quantify objects, and it can be relied upon because it is consistent and based on logic. Mathematics pervades the sciences and can be used to model planetary orbits, atomic motion, signal frequencies, and much more. These phenomena can be encoded with mathematics precisely and concisely. This has even led some to describe mathematics as being "unreasonably effective" (Wigner, 1960). These observations speak to the broad reach and domain-generality of mathematics.

In machine learning, mathematics is a valuable testbed for *problem-solving ability*: the ability to analyze a problem, pick out good heuristics from a large set of possibilities, and chain them together to produce an answer. This contrasts with plug-and-chug calculations, a skill which ML models can already exhibit (Henighan et al., 2020). Visual or linguistic reasoning may involve limited problem-solving ability for tasks such as image classification, but unlike math this is not the focus of these domains.

To measure the problem-solving ability of machine learning models, we introduce the MATH dataset, which consists of $12,500$ problems from high school math competitions. Given a problem from MATH, machine learning models generate a sequence, such as `$\frac{2}{3}$`, that encodes the final answer. These answers are unique after normalization, allowing MATH to be scored with exact match rather than with heuristic metrics such as BLEU. In addition, MATH problems are tagged by difficulty from 1 to 5, and span seven subjects including geometry, where diagrams can be specified in text with the Asymptote language. This enables a fine-grained assessment of

35th Conference on Neural Information Processing Systems (NeurIPS 2021) Track on Datasets and Benchmarks.

### Metamath Theorem Proving

$n \in \mathbb{N} \land \frac{n+1}{2} \in \mathbb{N} \implies \exists m \in \mathbb{N} : n = 2m + 1$.
GPT-$f$'s generated proof:

```
|- ((N e. NN0 /\ ((N + 1)/2) e.
   NN0) -> ((N - 1) / 2) e. NN0)
|- (N e. NN0 -> N e. CC)
|- 1 e. CC
|- ((N e. CC /\ 1 e. CC) ->
   (N - 1) e. CC )
                          ⋮
```

### DeepMind Mathematics Dataset

```
Divide 1136975704 by -142121963.
A: -8
Let k(u) = u**2+u-4. Find k(0).
A: -4
Sort 2, 4, 0, 6.
A: 0, 2, 4, 6
Solve 4 - 4 - 4 = 188*m for m.
A: -1/47
```

### MATH Dataset (Ours)

**Problem:** Tom has a red marble, a green marble, a blue marble, and three identical yellow marbles. How many different groups of two marbles can Tom choose?

**Solution:** There are two cases here: either Tom chooses two yellow marbles (1 result), or he chooses two marbles of different colors ($\binom{4}{2} = 6$ results). The total number of distinct pairs of marbles Tom can choose is $1 + 6 = \boxed{7}$.

**Problem:** The equation $x^2 + 2x = i$ has two complex solutions. Determine the product of their real parts.

**Solution:** Complete the square by adding 1 to each side. Then $(x + 1)^2 = 1 + i = e^{\frac{i\pi}{4}}\sqrt{2}$, so $x + 1 = \pm e^{\frac{i\pi}{8}}\sqrt[4]{2}$. The desired product is then $\left(-1 + \cos\left(\frac{\pi}{8}\right)\sqrt[4]{2}\right)\left(-1 - \cos\left(\frac{\pi}{8}\right)\sqrt[4]{2}\right) = 1 - \cos^2\left(\frac{\pi}{8}\right)\sqrt{2} = 1 - \frac{\left(1+\cos\left(\frac{\pi}{4}\right)\right)}{2}\sqrt{2} = \boxed{\dfrac{1 - \sqrt{2}}{2}}$.

Figure 1: Previous work is based on formal theorem provers or straightforward plug-and-chug problems. Our dataset, MATH, has competition mathematics problems with step-by-step solutions written in LaTeX and natural language. Models are tasked with generating tokens to construct the final (boxed) answer.

mathematical problem-solving ability across difficulties and subjects. Finally, problems come with full step-by-step solutions. By training on these, models can learn to generate their own step-by-step solutions, which can facilitate learning and make model outputs more interpretable.

The MATH dataset is challenging: large language models achieved accuracies ranging from $3.0\%$ to $6.9\%$. Despite these low accuracies, models clearly possess some mathematical knowledge: they achieve up to $15\%$ accuracy on the easiest difficulty level, and they are able to generate step-by-step solutions that are coherent and on-topic even when incorrect. We also evaluated humans on MATH, and found that a computer science PhD student who does not especially like mathematics attained approximately $40\%$ on MATH, while a three-time IMO gold medalist attained $90\%$, indicating that MATH can be challenging for humans as well.

The presence of step-by-step solutions allows models to utilize "scratch space": rather than having to generate a final answer immediately, models can first generate solutions that may contain intermediate computations. Interestingly, we found that having models generate step-by-step solutions before producing an answer actually *decreased* accuracy relative to immediately outputting a final answer without generating solutions, indicating the solutions are currently not useful for models at test time. In contrast, having models *train* on solutions increases relative accuracy by $10\%$ compared to training on the questions and answers directly. We also find that models do better with hints in the form of partial solutions. Our results show that models can make use of actual step-by-step solutions provided to them in various ways, but that they are still unable to effectively use their own generated solutions. Bridging this gap poses an interesting direction for further research.

While MATH covers advanced problem-solving techniques, models may first need to be trained thoroughly on the fundamentals of mathematics. To address this, we create the first large-scale mathematics pretraining dataset with hundreds of thousands of step-by-step solutions in natural language and LaTeX. We call this dataset the Auxiliary Mathematics Problems and Solutions (AMPS) pretraining corpus, which consists of Khan Academy and Mathematica data. AMPS has over $100,000$ Khan Academy problems with step-by-step solutions in LaTeX; these exercises are used to teach human students concepts ranging from basic addition to Stokes' Theorem. It also contains over 5 million problems generated using Mathematica scripts, based on 100 hand-designed modules covering topics such as conic sections, div grad and curl, KL divergence, eigenvalues, polyhedra, and Diophantine equations. In total AMPS contains 23GB of problems and solutions. Pretraining on

AMPS enables a 0.1 billion parameter model to perform comparably to a fine-tuned model that is $130\times$ larger.

Altogether, while large Transformer models (Vaswani et al., 2017) make some progress on the MATH dataset, such as by AMPS pretraining or by training with step-by-step solutions, accuracy nonetheless remains relatively low. While enormous Transformers pretrained on massive datasets can now solve most existing text-based tasks, this low accuracy indicates that our MATH dataset is distinctly harder. Accuracy also increases only modestly with model size: assuming a log-linear scaling trend, models would need around $10^{35}$ parameters to achieve $40\%$ accuracy on MATH, which is impractical. Instead, to make large strides on the MATH dataset with a practical amount of resources, we will need new algorithmic advancements from the broader research community.

## 2  Related Work

**Neural Theorem Provers.**  Much of the existing work on machine learning models for mathematical reasoning relies on automated theorem proving benchmarks. Huang et al. (2019) use the Coq theorem proving environment to create a machine learning benchmark with $1,602$ theorems and lemmas. Bansal et al. (2019) introduce the HOList benchmark for automated theorem proving, which uses a formal language to enable automatic evaluation. Rather than use HOList, Polu and Sutskever (2020) use the Metamath formalization language for automated theorem proving with promising results. We show an example of Metamath in Figure 1. These benchmarks can be approached with seq2seq (Sutskever et al., 2014) Transformers which have traction on the problem (Polu and Sutskever, 2020; Rabe et al., 2020; Li et al., 2020).

Rather than prove theorems with standard pretrained Transformers, McAllester (2020) proposes that the community create theorem provers that bootstrap their mathematical capabilities through open-ended self-improvement. For bootstrapping to be feasible, models will also need to understand mathematics as humans write it, as manually converting advanced mathematics to a proof generation language is extremely time-consuming. This is why Szegedy (2020) argues that working on formal theorem provers alone will be an impractical path towards world-class mathematical reasoning. We address Szegedy (2020)'s concern by creating a dataset to test understanding of mathematics written in natural language and commonplace mathematical notation. This also means that the answers in our dataset can be assessed without the need for a cumbersome theorem proving environment, which is another advantage of our evaluation framework.

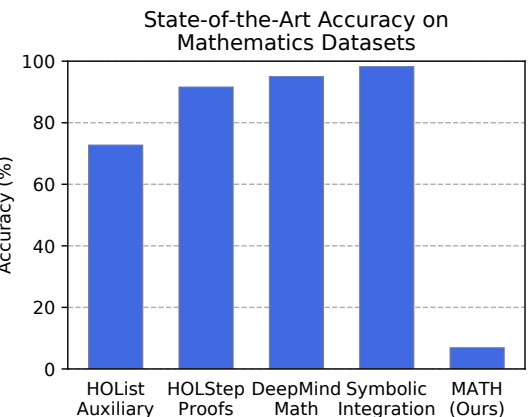

Figure 2: Compared to existing proof and plug-and-chug tasks, our mathematical problem solving task is considerably more challenging. HOList results are from Wu et al. (2021). HOLStep results are from Crouse et al. (2019). DeepMind Math accuracy is the median IID accuracy from Henighan et al. (2020). Symbolic Integration accuracy is from Lample and Charton (2020).

**Neural Calculators.**  Recent work shows that Transformers can sometimes perform laborious calculations around as well as calculators and computer algebra systems. Lample and Charton (2020) use Transformers to solve algorithmically generated symbolic integration problems and achieve greater than 95% accuracy. Amini et al. (2019); Ling et al. (2017) introduce plug-and-chug multiple choice mathematics problems and focus on sequence-to-program generation. Saxton et al. (2019) introduce the DeepMind Mathematics dataset, which consists of algorithmically generated plug-and-chug problems such as addition, list sorting, and function evaluation, as shown in Figure 1. Recently, Henighan et al. (2020) show that, excluding problems with astronomically large numbers, the vast majority of the problems in the DeepMind Mathematics dataset can be straightforwardly solved with large Transformers.

**Benchmarks for Enormous Transformers.**  There are few existing natural language benchmarks left to solve, as tasks that aggregate multiple subtasks such as SuperGLUE (Wang et al., 2019) are solved by simply training enormous Transformers (He et al., 2020). Kaplan et al. (2020); Henighan

| | |
|---|---|
| Algebra | Conic sections, polynomial GCD, De Moivre's theorem, function inverses, ... |
| Calculus | Arclength, Jacobian, Laplacian, divergence, curl, gradients, integrals, ... |
| Statistics | Expectation, geometric mean, harmonic mean, KL divergence, variance, ... |
| Geometry | Triangle area, triangle inradius, polygon angles, polyhedron diameter, ... |
| Linear Algebra | Characteristic polynomials, eigenvalues, reduced row echelon form, ... |
| Number Theory | Modular inverse, Euler's totient function, Chinese remainder theorem, ... |

Table 1: A subset of the topics covered by our 100 hand-designed Mathematica scripts, which is part of our Auxiliary Mathematics Problems and Solutions (AMPS) pretraining dataset. Of these scripts, 37 also generate step-by-step solutions. We generated around 50,000 exercises with each Mathematica script, or around 5 million problems.

et al. (2020) show that the performance of Transformers predictably increases with an increase in model size and dataset size, raising the question of whether natural language processing can be solved by simply increasing compute and funding. Additionally, Chen et al. (2021); Austin et al. (2021) show that code generation models scale reliably across several orders of magnitude, and, should scaling continue, Chen et al. (2021)'s HumanEval code generation dataset should be solved in a few orders of magnitude. In the Supplementary Materials, we even find that large GPT-3 models can perform remarkably well on a sequence completion test similar to an IQ test, the C-Test (Hernández-Orallo, 1998; Legg and Hutter, 2007). Even difficult logical understanding tasks such as LogiQA (Liu et al., 2020) will soon be straightforwardly solved by enormous Transformers should trends continue, which we also show in the Supplementary Materials. Hendrycks et al. (2021) create a multiple-choice benchmark covering 57 subjects. However, unlike our benchmark, which is a text generation task with 12, 500 mathematical reasoning questions, their benchmark is a multiple choice task that includes only a few hundred questions about mathematics. In contrast to these benchmarks, we find that our MATH benchmark is unusually challenging for current models and, if trends continue, simply using bigger versions of today's Transformers will not solve our task in the foreseeable future.

## 3 Datasets

In this section, we introduce two new datasets, one for benchmarking mathematical problem-solving ability (MATH) and one for pretraining (AMPS).

### 3.1 The MATH Dataset

The MATH dataset consists of problems from mathematics competitions including the AMC 10, AMC 12, AIME, and more. Many of these problems can be collected from aops.com/community/c3158_usa_contests. These competitions span decades and assess the mathematical problem-solving ability of the best young mathematical talent in the United States. Unlike most prior work, most problems in MATH cannot be solved with a straightforward application of standard K-12 mathematics tools. Instead, humans often solve such problem by applying problem solving techniques and "heuristics" (Pólya, 1945).

The Mathematics Aptitude Test of Heuristics dataset, abbreviated MATH, has 12,500 problems (7,500 training and 5,000 test). With this many training problems, models can learn many useful heuristics for problem solving. Each problem has a step-by-step solution and a final boxed answer. Example problems with step-by-step solutions are shown in Figure 1.

**Categorizing Problems.** Problems span various subjects and difficulties. The seven subjects are Prealgebra, Algebra, Number Theory, Counting and Probability, Geometry, Intermediate Algebra, and Precalculus. While subjects like Prealgebra are generally easier than Precalculus, within a subject problems can take on different difficulty levels. We encode a problem's difficulty level from '1' to '5,' following AoPS. A subject's easiest problems for humans are assigned a difficulty level of '1,' and a subject's hardest problems are assigned a difficulty level of '5.' Concretely, the first few problems of an AMC 8 exam are often level 1, while AIME problems are level 5. This allows us to assess performance across both different subjects and different levels of difficulty.

**Formatting.** Problems and solutions are consistently formatted using LaTeX and the Asymptote vector graphics language. Our usage of LaTeX allows us to flexibly encode mathematical problems while avoiding unusual symbols or cumbersome formal languages. Meanwhile, mathematical figures are encoded in the Asymptote language rather than as raster images. This enables pure language

| Model | Prealgebra | Algebra | Number Theory | Counting & Probability | Geometry | Intermediate Algebra | Precalculus | Average |
|---|---|---|---|---|---|---|---|---|
| GPT-2 0.1B | 5.2 | 5.1 | 5.0 | 2.8 | 5.7 | 6.5 | 7.3 | 5.4 +0% |
| GPT-2 0.3B | 6.7 | 6.6 | 5.5 | 3.8 | 6.9 | 6.0 | 7.1 | 6.2 +15% |
| GPT-2 0.7B | 6.9 | 6.1 | 5.5 | 5.1 | 8.2 | 5.8 | 7.7 | 6.4 +19% |
| GPT-2 1.5B | 8.3 | 6.2 | 4.8 | 5.4 | 8.7 | 6.1 | 8.8 | 6.9 +28% |
| GPT-3 13B* | 4.1 | 2.4 | 3.3 | 4.5 | 1.0 | 3.2 | 2.0 | 3.0 −44% |
| GPT-3 13B | 6.8 | 5.3 | 5.5 | 4.1 | 7.1 | 4.7 | 5.8 | 5.6 +4% |
| GPT-3 175B* | 7.7 | 6.0 | 4.4 | 4.7 | 3.1 | 4.4 | 4.0 | 5.2 −4% |

Table 2: MATH accuracies across subjects. '*' indicates that the model is a few-shot model. The character 'B' denotes the number of parameters in billions. The gray text indicates the *relative* improvement over the 0.1B baseline. All GPT-2 models pretrain on AMPS, and all values are percentages. GPT-3 models do not pretrain on AMPS due to API limits. Model accuracy is increasing very slowly, so much future research is needed.

models to process figures, diagrams, and graphics, making it possible to assess these models on subjects such as geometry for the first time.

To assess models using exact match, we force the final boxed answers to follow consistent formatting rules. Specifically, probabilities are expressed as simplified fractions. Moreover, matrix entry fractions are encoded with `x/y`, while all other fractions are consistently encoded with the `\frac{x}{y}` command. Coefficients are encoded without a multiplication symbol (e.g. `5x` not `5*x`). Expressions with multiple variables are entered in alphabetical order; polynomials are expressed in decreasing degree order. Different fraction encodings equivalent, such as `\frac{x}{y}` and `\dfrac{x}{y}` and `x/y`. Different parenthesis encodings, such as `\left (` and `(`, are treated as equivalent.

We also allow units to be included or omitted from an answer, we ignore spaces, and we treat common equivalent ways of expressing the same number (e.g., $0.5$ and $1/2$, or $0.1$ and $.1$) as the same. When the answer is a factorized polynomial, we permit different orderings of the factors, so that $4(x + 1)(x − 1)$ is equivalent to $4(x − 1)(x + 1)$, and so on. These rules cover nearly all ways that different generated or actual solutions can be equivalent in practice.

**Automatically Assessing Generated Answers.** Due to design choices in MATH, we can assess the answers generated by a model *automatically*, even though the space of model outputs is combinatorially large. Automatic assessment starts by determining the beginning and end of the answer. This is possible to do even if a model generates step-by-step solutions because the final answers in MATH are wrapped and delimited with the `\boxed{}` command. We can consequently evaluate a model's output by parsing what is inside the `\boxed{}` command and comparing that with the ground truth answer, while accounting for the equivalent ways of formatting a string described above. Together, the box delimiter and formatting rules provide a unique answer in a well-defined location, which allows us to test for equivalence and use accuracy as our primary metric.

**Human-Level Performance.** To provide a rough but informative comparison to human-level performance, we randomly sampled 20 problems from the MATH test set and gave them to humans. We artificially require that the participants have 1 hour to work on the problems and must perform calculations by hand. All participants are university students. One participant who does not like mathematics got $8/20 = 40\%$ correct. A participant ambivalent toward mathematics got $13/20$. Two participants who like mathematics got $14/20$ and $15/20$. A participant who got a perfect score on the AMC 10 exam and attended USAMO several times got $18/20$. A three-time IMO gold medalist got $18/20 = 90\%$, though missed questions were exclusively due to small errors of arithmetic. Expert-level performance is theoretically $100\%$ given enough time. Even $40\%$ would accuracy for a machine learning model would be impressive but have ramifications for cheating on homework.

## 3.2 AMPS (Khan + Mathematica) Dataset

Since pretraining data can greatly influence performance (Hernandez et al., 2021; Gururangan et al., 2020) and since mathematics is a small fraction of online text, we introduce a large and diverse mathematics pretraining corpus. Our pretraining dataset, the Auxiliary Mathematics Problems and Solutions (AMPS) dataset, has problems and step-by-step solutions typeset in LaTeX. AMPS contains over 100,000 problems pulled from Khan Academy and approximately 5 million problems generated from manually designed Mathematica scripts.

**Khan Academy.** The Khan Academy subset of AMPS has 693 exercise types with over 100,000 problems and full solutions. Problem types range from elementary mathematics (e.g. addition) to multivariable calculus (e.g. Stokes' theorem), and are used to teach actual K-12 students. The exercises can be regenerated using code from github.com/Khan/khan-exercises/. We show the full list of problem types in the Supplementary Materials.

**Mathematica.** To make AMPS larger, we also contribute our own Mathematica scripts to generate approximately $50\times$ more problems than our Khan Academy dataset. With Mathematica, we designed 100 scripts that test distinct mathematics concepts, 37 of which include full step-by-step LaTeX solutions in addition to final answers. We generated around 50,000 exercises from each of our scripts, or around 5 million problems in total. This results in over 23 GB of mathematics problems, making it larger than the 16 GB of natural language used to train BERT (Devlin et al., 2019).

Problems include various aspects of algebra, calculus, counting and statistics, geometry, linear algebra, and number theory (see Table 1 for a sampling of topics). Unlike prior approaches to algorithmically generating mathematics problems, we use Mathematica's computer algebra system so that we can manipulate fractions, transcendental numbers, and analytic functions.

## 4 Experiments

In this section, we perform experiments to investigate performance on the MATH dataset. We find that accuracy remains low even for the best models. Furthermore, unlike for most other text-based datasets, we find that accuracy is increasing very slowly with model size. If trends continue, then we will need algorithmic improvements, rather than just scale, to make substantial progress on MATH. Nevertheless, we show that making progress is also possible today. We find that pretraining on AMPS enables a small 0.1B parameter model to perform similarly to a large fine-tuned 13B parameter model.

We also experiment with using step-by-step solutions. We find that having models generate their own step-by-step solutions before producing an answer actually *degrades* accuracy. We qualitatively assess these generated solutions and find that while many steps remain illogical, they are often related to the question. Finally, we show that step-by-step solutions can still provide benefits today. We find that providing partial ground truth step-by-step solutions can improve performance, and that providing models with step-by-step solutions at training time also increases accuracy.

### 4.1 Experimental Setup

**Models and Hyperparameters.** Because MATH answers must be generated, we use autoregressive language models, namely GPT-2 (Radford et al., 2016) and GPT-3 (Brown et al., 2020), which are decoder models pretrained on natural language text. Our GPT-2 models tokenizes numbers so that one digit is processed at a time (Henighan et al., 2020). T5's (Raffel et al., 2020) tokenizer removes many LaTeX symbols, so after a broad hyperparameter sweep lasting two weeks, its performance was not competitive. We show results with the BART architecture in the Appendix.

Before fine-tuning on MATH, models pretrain on AMPS. We pretrain for one epoch, using AdamW (Loshchilov and Hutter, 2019), using a batch size of $128$, and using a weight decay of $0.05$. We use the standard autoregressive language modeling objective. During pretraining, we upsample Khan Academy data by a factor of 5 and we downsample Mathematica by a factor of 2 to account for the large difference in dataset sizes.

During fine-tuning, models predict final answers and solutions. Concretely, if $\langle P \rangle$ is the problem statement, we train with an equal mix of "$\langle P \rangle$ `Final Answer: <Answer>`" and "$\langle P \rangle$ `Full Solution: <Step-by-Step Solution>`" sequences. This makes it possible for the model to both generate full solutions and also to output just the final answer. For fine-tuning we use the same batch size and weight decay as in pretraining. Models are trained with 8 A100 GPUs, each requiring less than a day.

Unless otherwise specified, for GPT-2 we use the default HuggingFace (Wolf et al., 2020) generation parameters, except that we use beam search. Our beam search has a beam size of 20 when only generating the final answer, and a beam size of 10 when generating full step-by-step solutions. By default, we evaluate models by prompting them with "$\langle P \rangle$ `Final Answer:`" so that they directly generate the final answer to each problem, not the step-by-step solution.

**Problem:** Suppose $a$ and $b$ are positive real numbers with $a > b$ and $ab = 8$. Find the minimum value of $\frac{a^2+b^2}{a-b}$.

**Generated solution:** Let $s = a + b$. By QM-AM, $\sqrt{\frac{a^2+b^2}{a-b}} \geq \frac{a+b}{2} = \frac{s}{2}$. Then $\frac{a^2+b^2}{a-b} \geq \frac{s^2}{4}$, so $a^2 + b^2 \geq \frac{s^2}{2}$. Hence, $\frac{a^2+b^2}{a-b} \geq \frac{s^2}{2} + \frac{2s^2}{a-b} = \frac{s^2}{2} + 2s^2 = \boxed{8}$.

**Ground truth solution:** We can write $\frac{a^2+b^2}{a-b} = \frac{a^2+b^2-2ab+16}{a-b} = \frac{(a-b)^2+16}{a-b} = a - b + \frac{16}{a-b}$. By AM-GM, $a - b + \frac{16}{a-b} \geq 2\sqrt{(a-b) \cdot \frac{16}{a-b}} = 8$. Equality occurs when $a - b = 4$ and $ab = 8$. We can solve these equations to find $a = 2\sqrt{3} + 2$ and $b = 2\sqrt{3} - 2$. Thus, the minimum value is $\boxed{8}$.

---

**Problem:** Right $\triangle ABC$ has legs measuring 8 cm and 15 cm. The triangle is rotated about one of its legs. What is the number of cubic centimeters in the maximum possible volume of the resulting solid? Express your answer in terms of $\pi$.

**Generated solution:** Rotating the triangle about one leg gives us the following triangle:

$$\frac{(8)(15)}{(8)(-1) + (15)(-1)} = \frac{120}{7} = \boxed{120\pi}.$$

**Ground truth solution:** If the triangle is rotated about the shorter leg, then the radius is the longer leg and the height is the shorter leg, and the volume is $\frac{1}{3} \cdot (15^2\pi)(8) = 600\pi$ cubic centimeters. If the triangle is rotated about the longer leg, then the radius is the shorter leg and the height is the longer leg, and the volume is $\frac{1}{3}(8^2\pi)(15)$, which is $\frac{8}{15}$ of the volume we found earlier. So, the maximum possible volume is $\boxed{600\pi}$ cubic centimeters.

Figure 3: Problems, step-by-step solutions generated by our GPT-2 1.5B model, and ground truth solutions. Observe that models can provide the right answer yet generate a misleading and wrong explanation. The second generated solution demonstrates that models are capable of generating Asymptote commands to create figures and graphics.

We also evaluate GPT-3 with fine-tuning and also in a few-shot setting using the OpenAI API. We use the 'Curie' GPT-3 model which has approximately 13 billion parameters, and the 'Davinci' model which has approximately 175 billion parameters. When performing few-shot evaluation, we construct our prompt by prepending 8 problems with correct answers (but not step-by-step solutions due to space). Using temperature 0, models output up to 20 tokens for the final answer. The OpenAI API also allows users to fine-tune models up to 13B parameters at the time of writing, but their API does not have the option to pretrain on datasets as large as AMPS.

## 4.2 Analyzing Model Performance

**AMPS Pretraining.** As an ablation, we test how models with AMPS pretraining compare with models that were not pretrained on AMPS. Without pretraining on AMPS, a GPT-3 (13B) model fine-tuned on MATH attains 5.2% accuracy. In contrast, a GPT-2 (0.1B) model both pretrained on AMPS and fine-tuned on MATH attains 5.4%. Consequently AMPS increases accuracy about as much as a $130\times$ increase in parameters, demonstrating its value as a pretraining dataset.

We additionally tried pretraining on StackExchange, a real-world but less curated source of mathematics text. A GPT-2 (0.3B) model pretrained on both AMPS and questions and answers from Math StackExchange ($\sim 3$ GB) had 6.0% accuracy. This is actually less than the 6.2% accuracy attained by pretraining on AMPS alone. Thus our dataset is more useful for pretraining even than diverse real-world mathematics data.

**Model Size.** While increasing model parameters often automatically solves many tasks (Brown et al., 2020), we find that MATH is unusually challenging for enormous Transformers. Table 2 shows that the average accuracy across subjects for the smallest model, GPT-2 with 0.1 billion parameters, is 5.4%. Meanwhile, a GPT-2 model with $15\times$ the number of parameters attains 6.9% accuracy, a 28% relative improvement. This indicates that while having more parameters helps, absolute accuracy remains far from the ceiling and is only increasing slowly, quite unlike most other text-based tasks.

**Problem:** The three-digit number "$ab5$" is divisible by 3. How many different three-digit numbers can "$ab5$" represent?

**Generated solution:** A number is divisible by 3 if and only if the sum of its digits is divisible by 3. So $ab5''$ is divisible by 3 if and only if the two-digit number $ab$ leaves a remainder of 1 when divided by 3. There are 90 two-digit numbers, of which $90/3 = \boxed{30}$ leave a remainder of 1 when divided by 3.

**Ground truth solution:** If the number is divisible by 3, the sum of the digits must be divisible by 3. So $a + b$ must be one more than a multiple of 3, since 5 is one less than a multiple of 3. We have several options: If $a + b = 1$, it must be $(1, 0)$, one possibility. For $a + b = 4$, $a$ can be 1 to 4, for four possibilities. For $a + b = 7$, $a$ can be 1 to 7, so seven possibilities. For $a + b = 10$, anything from $(1, 9)$ to $(9, 1)$ works, so nine possibilities. For $a + b = 13$, the pairs range from $(4, 9)$ to $(9, 4)$, for six possibilities. If $a + b = 16$, we can have $(7, 9)$, $(8, 8)$, or $(9, 7)$, so three possibilities. $a + b = 19$ and up is not possible. So the total number is $1 + 4 + 7 + 9 + 6 + 3 = \boxed{30}$

---

**Problem:** Beginning at point $A$ in the diagram, Dora selects one of the four possible directions with equal probability. Each time she comes to an intersection, she again randomly selects one of the possible directions. What is the probability that, in her first four steps, she will walk completely around the gray square? Express your answer as a common fraction.

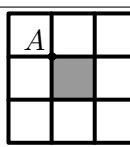

**Generated solution:** The total probability that Dora chooses a direction completely around the gray square is $\frac{5}{10} = \boxed{\frac{1}{2}}$.

**Ground truth solution:** The only way for the Dora to end up at her starting point in four steps is for her to traverse the four sides of the gray square. She can do this in two ways: clockwise and counterclockwise. The probability of each of these two paths is $\left(\frac{1}{4}\right)^4 = \frac{1}{256}$. Therefore, the probability that she ends up where she started is $\frac{1}{256} + \frac{1}{256} = \boxed{\frac{1}{128}}$.

Figure 4: Additional example problems, generated solutions, and ground truth solutions from our MATH dataset. The first problem's generated solution has the right answer with a correct and simple explanation. The second problem is a combinatorics problem specified with a figure, which the model gets wrong.

**Problem Difficulty.** We also analyze model accuracy while controlling for problem difficulty. Higher levels of difficulty correspond to lower accuracy, as expected. These results are visualized in the Supplementary Materials. The accuracy of GPT-2 (1.5B) is around $15\%$ for level 1 (easy) and around $4\%$ for level 5 (hard). Even our benchmark's easiest problems are more challenging than previous benchmarks that focused on straightforward plug-and-chug problems.

**Error Detection.** To determine whether we can trust the answers from a model, we analyze model confidence to see whether confidence tends to be higher for correct answers. We define confidence as the average prediction probability of the tokens that make up a generated answer. GPT-2 (1.5B) is highly overconfident, with confidences that are often around $100\%$. Moreover, there is substantial overlap between correct and incorrect answers. Following Hendrycks and Gimpel (2017), we computed the probability that a correct answer has higher confidence than an incorrect answer. To do this, we compute the Area Under the Receiver Operating Characteristic curve (AUROC). An AUROC of $100\%$ corresponds to being able to perfectly detect correct and incorrect answers, while $50\%$ corresponds to random chance. We find that with GPT-2 (1.5B), the AUROC is quite low at $68.8\%$. This suggests there is substantial room for improvement in detecting model errors.

### 4.3 Analyzing Step-by-Step Solutions

**Scratch Space.** Our MATH dataset and AMPS pretraining dataset provide full step-by-step solutions, an important and rare type of side information (Murty et al., 2020) that can in principle teach models how to derive answers and use scratch space. By training a language model on these solutions, we can have models generate full step-by-step solutions. This may be especially useful for difficult problems, for which outputting the correct answer after just a few forward passes may be insufficient. By allowing the model to use several steps of processing before outputting a final answer, the model could adaptively use computation and have higher performance, in addition to making its reasoning more interpretable.

We test this by prompting models with "⟨P⟩ Full Solution:" to generate a full solution along with a final boxed answer, rather than the boxed answer alone. We evaluated this for GPT-2 (1.5B) and found that this actually makes performance worse, dropping accuracy to 5.3%. We hypothesize that the drop in accuracy from using scratch space arises from a snowballing effect, in which partially generated "solutions" with mistakes can derail subsequent generated text. Nevertheless, when generation becomes more reliable and models no longer confuse themselves by their own generations, our dataset's solutions could in principle teach models to use scratch space and attain higher accuracy.

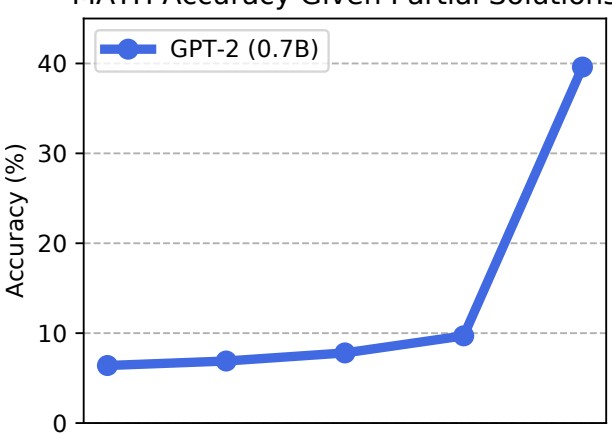

Figure 5: Models conditioned on most of a problem's step-by-step solution can often understand the solution to predict the final answer. Not all solutions have an answer that is immediate from the preceding solution text, though many are. '99%' of a solution is all the solution text before the final answer. This demonstrates that, even with substantial help, models are still struggling.

**Examples.** We can also qualitatively assess the step-by-step solutions that the model generates. We show examples of generated solutions in Figures 3 and 4. We find that the model can consistently generate correct LaTeX and often performs steps that appear related to the question at hand, but still makes many logical mistakes, both in terms of what the question seems to be asking and in individual steps that are part of a larger derivation.

**The Benefits of MATH Solutions.** We find that giving models partial step-by-step MATH solutions during inference can improve accuracy. We test performance when we allow models to predict the final answer given a "hint" in the form of a portion of the ground truth step-by-step solution. To do so, for this experiment we prompt models with "⟨P⟩ <Partial Step-by-Step Solution without Final Answer> Final Answer:" during both fine-tuning and evaluation for different partial fractions of the step-by-step solution. This is the same as the default setting when we let models see 0% of the step-by-step solution. When models see "99%" of the solution, they are given the whole step-by-step solution except for the final answer. We show results with GPT-2 (0.7B) for different fractions of the solution in Figure 5. Observe that the model still only attains approximately 40% when given 99% of the solution, indicating room for improvement.

Finally, we also find that providing models with step-by-step during training can further improve performance. We run an ablation by fine-tuning models on MATH with the same setup as before, except that we only show examples with the final answer and no step-by-step solution. If we fine-tune with only the final answer, the GPT-2 (1.5B) accuracy decreases by 0.6% to 6.3%.

## 5   Conclusion

In this paper, we laid groundwork for future research in machine learning for mathematical problem solving. We introduced the MATH benchmark, which enables the community to measure mathematical problem-solving ability. In addition to having answers, all MATH problems also include answer explanations, which models can learn from to generate their own step-by-step solutions. We also introduce AMPS, a diverse pretraining corpus that can enable future models to learn virtually all of K-12 mathematics. While most other text-based tasks are already nearly solved by enormous Transformers, MATH is notably different. We showed that accuracy is slowly increasing and, if trends continue, the community will need to discover conceptual and algorithmic breakthroughs to attain strong performance on MATH. Given the broad reach and applicability of mathematics, solving the MATH dataset with machine learning would be of profound practical and intellectual significance.

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
