# OpenReview forum: "Measuring Mathematical Problem Solving With the MATH Dataset"
_NeurIPS.cc/2021/Track/Datasets_and_Benchmarks/Round2 — NeurIPS 2021 Datasets and Benchmarks Track (Round 2)_

### Official Review · Reviewer_rA8v · 2021-09-14
**Really meaningful and challenging dataset for mathematical problem solving**

**Rating:** 9
**Confidence:** 4
**Clarity:** The paper is clear and easy to follow.

**Strengths:**

The MATH dataset is curated such that it contains not only question-answer pairs, but also a step-by-step solution for each question, expressed in natural language and symbolic formula.

The dataset is carefully formatted and normalized. All answers are expressed in a unique format so that evaluation doesn’t need to resort to fuzzy matching. Diagrams are also expressed in a unified scripting language.

**Weaknesses:**

The authors may consider setting up some benchmarks with non-fully-E2E-neural-models. For example, with methods that predict a numerical program and then use a symbolic calculator to give the final answer.

**Additional Feedback:**

It’d be great if the authors can release the mathematica scripts that are used to generate part of the AMPS pretraining dataset.

**Correctness:**

Per the description in the paper, the MATH and AMPS dataset are legitimately collected, parsed or generated. Experimental settings also look correct.

The MATH dataset is divided into a train set (7.5k examples) and a test set (5k examples), but there seems to be no development set.

**Documentation:**

The code and data are accessible via the link to GitHub repo. Code is moderately documented.

**Ethics:**

The potential negative societal impacts are well discussed in the paper.

**Relation To Prior Work:**

I like the extensive discussion of prior work on mathematical reasoning through theorem proving, and how the authors present MATH as providing an easy and objective way to measure progress in mathematical reasoning.

Yet MATH is a QA-style dataset, and there is not too much focus on its distinctiveness from other mathematical QA datasets, such as MAWPS, MathQA, ASDiv, SVAMP, etc. I would love to see more discussion in this area.

**Summary And Contributions:**

This paper presents the MATH dataset that poses mathematical problem-solving challenges with high-school competition-level questions. It also curates a larger math problem-solving corpus called AMPS with which we can do secondary pretraining.

Benchmarks show that MATH is extremely challenging for large language models, and that if we extrapolate the scaling trends, it would not be feasible to solve MATH by simply increasing the model size, thus encouraging algorithmic innovations in this domain.

---

### Official Review · Reviewer_ZVzu · 2021-09-21
**Measuring Mathematical Problem Solving With the MATH Dataset**

**Rating:** 6
**Confidence:** 4

**Strengths:**

The paper introduces two large datasets. The MATH dataset for benchmarking mathematical problem-solving of ability and the AMPS dataset for pretraining models. The problems in the datasets can also be categorized to different subject areas such as calculus, statistics, and algebra. Authors have shown that the dataset covers several subtopics from each of the subject areas. Moreover, the paper also uses a standard difficulty level of these math problems.
A comprehensive evaluation protocol that enables to evaluate models step by step. Each problem in the MATH dataset consists of a step-by-step solution and a final answer. This setup helps in evaluating models by varying the fraction of the solution seen by the model. Moreover, having different subject areas, difficulty level for each math problem enables a thorough analysis of model performance.


**Weaknesses:**

1.	The claims made by the paper seems too optimistic. Authors main goal is to introduce the MATH dataset to measure problem-solving ability of machine learning models. However, solving the problems of the dataset, does it really measure the ability of the models. In line 40 authors further mention that despite the low accuracies produced by the models, they clearly possess some mathematical knowledge. However, it’s not clear what evidence authors use to make these claims. In my opinion, the paper should restate the claims made.

2.	The design of the dataset is not entirely clear to me. Authors have identified different subject areas for the problems in the dataset, which is a plus point. However, it is not clear how the problems are distributed in the train and test sets. For example, if statistics problems are included in the test sets, are statistics problems included in the training set. If it cannot be ensured that problems from the same subject area is not included in the training, it is a major drawback in the dataset. Even human would not be able to solve problems in a certain subject area without learning the subject.

3.	It is not clear if the GPT-3 model is pretrained or not. In line 239 authors mention that models are pre-trained on AMPS and in Table 2 indicates that GPT-3 models do not pretrain. Please adjust this confusion in the paper. If the GPT-3 models are not pre-trained, could the authors explain why two different setting are used for GPT-2 and GPT-3 models?

4.	Authors have conducted an experiment with humans have an informative comparison between models and human performance, which is an interesting comparison to me. Authors have not mentioned how many human participants were present in this study.


**Additional Feedback:**

no

**Clarity:**

I think the paper could have been written in a more clear way. For example, the claims that authors made, the design of the dataset, and details of the models could have been discussed and justified in a comprehensible way.

**Correctness:**

Minor typos.
Line 221: accuracy is increasingly very slowly -> increasing
Caption in Table 2: on AMPS.Model a -> AMPS. Model


**Documentation:**

URL to the dataset is provided. Missing details about how to perform different evaluation settings discussed in the paper.

**Ethics:**

The paper has conducted a study with human participants. However, the checklist section 5 indicates as N/A. This could be directed to the section where the authors discuss about the human level performance.

**Relation To Prior Work:**

The papers discusses related work in detail from different areas.

**Summary And Contributions:**

This paper introduces the MATH dataset that includes challenging competition mathematics problems. The authors argue they measure the mathematical problem-solving ability of machine learning models using this dataset. Each problem in the dataset has a step-by-step solution, which enables to perform evaluations in multiple levels by showing different fractions of the solution to the model. The benchmark also includes a large auxiliary pretraining dataset. Experiments have done using different GPT-2 and GPT-3 models and results suggest that accuracy of those models are low on the dataset.

---

> ### Comment · Reviewer_ZVzu · 2021-10-02
> **Some concerns remaining**
>
> I thank the authors for clarifying some of the issues raised. My concern was that knowing how to solve “reduced row echelon form” problems in the training set is not enough to actually solve “eigenvalues” in the testing set. I recommend the authors include a table with the distribution of the instances for each of the topics covered (at least in the appendix). Are those questions sampled using the stratified sampling method? Also, the authors have not addressed my concern regarding the human evaluation. Please include complete human evaluation details in the revised version of the paper.
>
> Given that the authors have addressed the issues regarding agents and the contribution of the paper, I will raise my score to 6.

---

> > ### Author Response · Authors · 2021-10-02
> > **Reply**
> >
> > Thank you for your reply.
> >
> > > My concern was that knowing how to solve “reduced row echelon form” problems in the training set is not enough to actually solve “eigenvalues” in the testing set.
> >
> > For AMPS, models just pretrain on the whole set. MATH has a train test split, but AMPS is designed for pretraining and thus has no split. Consequently, models see both reduced row echelon form questions and eigenvalues questions during pretraining.
> >
> > > I recommend the authors include a table with the distribution of the instances for each of the topics covered (at least in the appendix).
> >
> > Since they are algorithmically generated, individual AMPS and Khan exercises are approximately balanced. Following your request, we intend to include more of this information, though it might take us a while to create a series of tables with statistics for all 600+ exercise types.
> >
> >  > Authors have not mentioned how many human participants were present in this study.
> >
> > Because MATH requires a strong mathematical background to perform well on, and a long amount of time to solve problems, we were restricted to assessing six human participants; we provide results for all of these participants are presented at the end of 3.1. We did not know of many three time IMO gold medalists so it was harder to find experts at this level. This was mainly to give a rough sense of the dataset's difficulty; many benchmark papers such as ImageNet do not include such studies.

---

### Official Review · Reviewer_uqAL · 2021-09-21
**Well motivated and well designed dataset that is an exciting and unsolved challenge**

**Rating:** 8
**Confidence:** 4
**Correctness:** Correctness does not appear to be an …
**Clarity:** This paper is very clearly written.

**Strengths:**

* well motivated as an important problem for AI to solve
* provides different levels of difficulty as well as a separate pre-training dataset, which enable investigating questions about bootstrapping, transfer learning, and curriculum learning
* well-formatted and easy-to-use dataset, being both easily human readable (mathematics is expressed in LaTeX), normalized (a unique answer per question, enabling automatic evaluation), and a language for diagram/figure representation (so that sequence models can parse them as well)
* interesting results already, such as the
  - large improvements on MATH dataset after pretraining on the AMPS dataset
  - decreases performance when models were forced to generate intermediate computations before generating a final answer
  - scaling analyses on the predicted performance of large-scale transformer models


**Weaknesses:**

- Perhaps it is a bit unfair to evaluate existing language models out-of-the-box on this dataset, especially using pre-existing tokenizers that are not designed for the particular statistics of the mathematics data (nor the graphics representations). As the paper mentions, for example, the T5 tokenizer removes many LaTeX symbols.
- It would be great to have some suggestions on how to do evaluation of the intermediate computations, even if this must be done on a manual basis (it seems that this was done to some extent)
- What is the accuracy of the language models on the AMPS dataset? This would give some sense of the difference in difficulty between the two datasets.


**Additional Feedback:**

--

**Documentation:**

There does not appear to be documentation on the following items: ethical and responsible use, intended uses; a hosting and maintenance plan

**Ethics:**

There is a mention that this dataset could enable cheating by students.


**Relation To Prior Work:**

This paper provides a challenge for large-scale transformer models that are much harder to solve than existing benchmarks on either language or mathematics tasks.

**Summary And Contributions:**

This paper presents the MATH dataset, consisting of problems from high school math contests. Unlike other math-related (or language) datasets, the MATH problems are currently unsolvable by large-scale transformer models, and scaling analyses indicate that they would not be solvable at any reasonable scale. Furthermore, mathematical problem solving is a key skill for artificial intelligence to solve. This paper additionally provides a supplementary dataset for pretraining, which they show can significantly improve performance on the MATH dataset.

In sum, this dataset presents an exciting challenge in that it could force creative innovation in ML architectures, while testing models on important dimensions like problem solving, reasoning, natural language understanding, explanation and interpretability, conditional computing, "working memory" / "scratch space" processing, and curriculum learning.

---

### Author Response · Authors · 2021-09-29
**Reviewer Responses**

We thank the reviewers for their careful analyses of our paper and their overall strong positive reception. We will respond point by point to each reviewer.

Reviewer uqAL,
1. “Tokenizers.” Some tokenizers, such as the tokenizer used by T5, are customized for natural language and are unsuitable for code and LaTeX. However, other tokenizers, such as those used by GPT and BART can output LaTeX. It appears that creating better tokenizers for mathematics is an active topic of research (e.g., https://arxiv.org/pdf/2102.13019.pdf). Since questions do not have astronomical numbers, existing tokenizers may be adequate, but it is worth investigating whether than can be better. We tried custom tokenizers for the paper, though we did not find any large improvements.
2. “AMPS.” Each task in AMPS should be approximately as difficult as those found in DM Math, on which Transformers already have  made substantial progress. (However, AMPS contains solutions and over an order of magnitude more problem types, making it more suitable for pretraining.)
3. “Partial Credit.” We agree a partial credit metric would be interesting, especially since models are doing poorly now. In the International Mathematical Olympiad, partial credit is given, but they do not have an objective process with clear-cut rules for determining points (indeed, the exact number of points to award a given solution is debated); this suggests there is not an existing objective automatic process for assigning partial credit for competition problems. Perhaps future ML models could be trained to score solution quality, or perhaps the log loss of the correct solution could be assessed.

Reviewer ZVzu,
1. “Too optimistic.” We think the paper is not unusually optimistic when we suggest that scaling current models with current losses will not solve the dataset. We agree that models can compute some arithmetic problems and so possess some “mathematical knowledge,” though we are primarily testing for whether models can apply this know-how to solve difficult problems. Consequently their ability to perform rote computations is not emphasized in MATH. We do not claim that our benchmark captures every aspect of “problem-solving ability”. Nevertheless, we believe it would be very impressive if models were to solve the benchmark, and that this does seem to require much stronger problem-solving abilities than is currently possible. By “possessing some mathematical knowledge”, we mean being able to solve some math problems correctly. While terms like “knowledge” are not well-defined (see Gettier problems), we do think it would require something like “mathematical knowledge” to be able to actually solve the MATH benchmark, and we were simply assessing whether there is some degree of traction in that direction.
2. “Train test split.” To clarify, we did a standard random train/test split, so we do have problems corresponding to each subject area in both the train and test sets.
3. “GPT-3 pretraining.” We are not able to pretrain GPT-3 since we are not from OpenAI. Fortunately, very recently, they enabled fine-tuning their 13B parameter model on research datasets. Thanks to your comment, we mention that this is a consequence of the API’s limitation. We also fix the typos you raised in the updated paper. If our comments addressed the thrust of some of your concerns, we kindly ask that you raise your score.

Reviewer rA8v,
1. “Calculator.” We wholly agree that model using tools such as calculators and Mathematica is an interesting and promising direction. Doing this well might currently require sequential decision making and reinforcement learning, but nonetheless this is a promising suggestion.
2. Further Comparisons. In the Supplementary Materials we have a section on “Expanded Dataset Comparisons.”
3. Scripts. We include the Mathematica scripts used to generate AMPS inside the AMPS tar file.

Again, thank you all for taking the time to review our work.

---

### Comment · Program_Chairs · 2021-10-13
**Official Ethical Review**

For dataset papers including some level of human participation, it is important to report the details of the group of involved participants. This includes the expertise level, and demographics of the involved humans, as well as information on the conditions and context of their contribution  (ie. payment, experimental set up details, etc.). If necessary, authors should submit the study to their institutional review board (IRB) process to see if they are exempt or need to take further measures.

---

> ### Author Response · Authors · 2021-10-13
> **Checklist Information Added**
>
> Thank you for carefully reviewing our paper. We have completed all of the requested information (the instructions: lines 76-80, the questions: lines 81-119, the expertise level and demographics: 68-71, exemption: line 73, and remuneration details: line 74) in the updated supplementary materials. We therefore complete the checklist and provide the requested details.

---

### Decision · Program_Chairs · 2021-10-09

**Decision:**

Accept

**Comment:**

This paper presents the MATH dataset that consists of math solving problems of high-school competition level. Overall, all reviewers acknowledge the significance and quality of the dataset as well as the newly-posed challenges. Although some concerns are also raised regarding the paper writing and clarification in some aspects, I think overall the paper provides enough contribution for the dataset track.

Flagged for an additional ethics review because this is a study with human participants, and the concerns raised by reviewers where not sufficiently addressed.